METHODS

# An approximate-copula distribution for statistical modeling

**Sarah S. Ji**[1]*, **Benjamin B. Chu**[2], **Hua Zhou**[1,3], **Kenneth Lange**[3,4,5]

**1** Department of Biostatistics, University of California, Los Angeles, Los Angeles, California, United States of America, **2** Department of Biomedical Data Science, Stanford University, Stanford, California, United States of America, **3** Department of Computational Medicine, University of California, Los Angeles, Los Angeles, California, United States of America, **4** Department of Human Genetics, University of California, Los Angeles, Los Angeles, California, United States of America, **5** Department of Statistics and Data Science, University of California, Los Angeles, Los Angeles, California, United States of America

* smji@g.ucla.edu

## Abstract

Copulas, generalized estimating equations, and generalized linear mixed models promote the analysis of grouped data where non-normal responses are correlated. Unfortunately, parameter estimation remains challenging in these three frameworks. Based on prior work of Tonda, we derive a new class of probability density functions that allow explicit calculation of moments, marginal and conditional distributions, and the score and observed information needed in maximum likelihood estimation. We also illustrate how the new distribution flexibly models longitudinal data following a non-Gaussian distribution. Finally, we conduct a tri-variate genome-wide association analysis on dichotomized systolic and diastolic blood pressure and body mass index data from the UK-Biobank, showcasing the modeling potential and computational scalability of the new distributional family.

## Author summary

Modeling correlated responses is computationally challenging beyond the Gaussian realm. For instance, how should repeated binary outcomes in longitudinal studies be modeled? When a dataset contains both continuous and discrete responses, how can their dependence be captured in a principled and efficient way? This paper introduces a new class of probability distributions that enables flexible modeling of correlated responses of mixed type. Inspired by statistical copulas, the proposed approach is designed to remain computationally efficient even in high-dimensional settings. We refer to this framework as an approximate copula model and show that it provides a promising alternative to classical methods such as generalized linear mixed models and generalized estimating equations. To demonstrate its flexibility and scalability, we apply the

**Data availability statement:** Software: https://github.com/OpenMendel/ApproxCopula.jl. Reproducibility: https://github.com/sarah-ji/ApproxCopula-reproducibility.

**Funding:** K.L. is supported by NIH GM141798. H.Z. is supported by NIH R01 DK142026, NSF DMS-2054253, NSF IIS-2205441, and NIH R35 GM141798. B.B.C. is supported by NIH R01MH113078, NIH R01MH123157, and NIH R56HG010812. S.S.J. is supported by NIH HG002536. The funders had no role in study design, data collection and analysis, decision to publish, or preparation of the manuscript.

approximate-copula model to genome-wide association (GWAS) data involving a mixture of continuous, binary, and count responses.

# 1 Introduction

## 1.1 Motivation

The analysis of correlated data is stymied by the lack of flexible multivariate distributions with fixed margins. Once one ventures beyond the confines of multivariate Gaussian distributions, analysis choices are limited. [8] launched the highly influential method of generalized estimating equations (GEEs). This advance allows generalized linear models (GLMs) to accommodate the correlated traits encountered in panel and longitudinal data and effectively broke the stranglehold of Gaussian distributions in analysis. The competing method of statistical copulas introduced earlier by Sklar is motivated by the same consideration [14]. Finally, generalized linear mixed models (GLMMs) [2,21] attacked the same problem. GLMMs are effective tools for modeling overdispersion and capturing the correlations of multivariate discrete data.

However, none of these three modeling approaches is a panacea. GEEs lack a well-defined likelihood, and estimation searches can fail to converge. For copula models, likelihoods exist, but are unwieldy, particularly for discrete outcomes. Copula calculations scale extremely poorly in high dimensions. Computing with GLMMs is problematic since their densities have no closed form and require evaluation of multidimensional integrals. Gaussian quadratures scale exponentially in the dimension of the parameter space. Markov Chain Monte Carlo (MCMC) can be harnessed in Bayesian versions of GLMMs, but even MCMC can be costly. For these reasons alone, it is worth pursuing alternative modeling approaches.

This brings us to an obscure paper by the Japanese mathematical statistician Tonda. Working within the framework of Gaussian copulas [15] and generalized linear models, Tonda introduces a device for relaxing independence assumptions while preserving computable likelihoods [17]. He succeeds brilliantly except for the presence of an annoying constraint on the parameter space of the new distribution class. The fact that his construction perturbs marginal distributions is forgivable.

## 1.2 Our contributions

The current paper has several purposes. First, by adopting a slightly different working definition, we show how to extend Tonda's construction to lift the awkward parameter constraint. Our new definition allows explicit calculation of (a) moments, (b) marginal and conditional distributions, and (c) the score and observed information of the log-likelihood and allows (d) generation of random deviates. Tonda tackles item (a), omits items (b) and (c), and mentions item (d) only in passing. For maximum likelihood estimation (MLE), he relies on a non-standard derivative-free algorithm [13] that scales poorly in high dimensions. We present two gradient-based algorithms for extracting high-dimensional MLEs.

The key computational advantage of this new definition is that the loglikelihood contain no determinants, matrix inverses, or multidimensional integrals, in contrast to other multivariate outcome models. These features resolve computational bottlenecks in parameter estimation. As a consequence, correlated but non-continuous responses can be efficiently analyzed. We advocate gradient-based estimation methods that avoid computationally intensive second derivatives. To provide asymptotic standard errors and confidence intervals, we capitalize on sandwich estimators. These rely on the observed information matrix computed after convergence. For completeness, we derive the exact Hessian for our approximate-copula loglikelihood.

Here we illustrate the flexibility of our model by closely studying two common scenarios: (1) longitudinal data analysis with non-Gaussian repeated measurements, and (2) multivariate analysis with mixtures of continuous, binary, and discrete outcomes. Scalable software for these techniques is either nonexistent or severely limited. Our simulation studies and real data examples highlight not only the virtues of the approximate-copula model but also its limitations. For reasons to be explained, we find that the model reflects reality best when the number of components of each independent sampling units is low or the correlations between responses within a unit are small.

### 1.3 Paper organization

In subsequent sections, we begin by introducing the approximate-copula model and studying its statistical properties. Then we illustrate how the model is used in practice to analyze correlated non-Gaussian variables. Getting the correlation structure of the variables right is a key step in modeling both longitudinal and multi-trait data. Next, we discuss the details of parameter estimation that enable model fitting. Finally, we present analysis results on both simulated and real data. These results showcase the speed and flexibility of the approximate-copula model. In particular, our genome-wide association (GWAS) example demonstrates its scalability.

## 2 Materials and methods

### 2.1 Notation

For the record, here are some notational conventions used in the sequel. All vectors and matrices appear in boldface. The entries of the vector $\mathbf{0}$ consist of 0's, and the standard basis vector $\mathbf{e}_i$ has all entries 0 except a 1 in entry $i$. The $\top$ superscript indicates a vector or matrix transpose. The Euclidean norm of a vector $\mathbf{x}$ is denoted by $\|\mathbf{x}\|$, and the spectral norm of a matrix $\mathbf{M} = (m_{ij})$ is $\|\mathbf{M}\| = \sup_{\mathbf{x} \neq \mathbf{0}} \frac{\|\mathbf{Mx}\|}{\|\mathbf{x}\|}$. For a smooth real-valued function $f(\mathbf{x})$, we write its gradient (column vector of partial derivatives) as $\nabla f(\mathbf{x})$, its first differential (row vector of partial derivatives) as $df(\mathbf{x}) = \nabla f(\mathbf{x})^\top$, and its second differential (Hessian matrix) as $d^2 f(\mathbf{x}) = d\nabla f(\mathbf{x})$. If $g(\mathbf{x})$ is vector-valued with $i$th component $g_i(\mathbf{x})$, then the differential (Jacobi matrix) $dg(\mathbf{x})$ has $i$th row $dg_i(\mathbf{x})$. The transpose $dg(\mathbf{x})^\top$ is the gradient of $g(\mathbf{x})$. Differentials $dg(\mathbf{x})$ can be constructed from directional derivatives $d_\mathbf{v} g(\mathbf{x}) = \lim_{t \to 0} \frac{g(\mathbf{x}+t\mathbf{v})-g(\mathbf{x})}{t}$.

### 2.2 Definition of the approximate-copula model

Consider $d$ independent random variables $X_1, \ldots, X_d$ with densities $f_i(x_i)$ relative to measures $\alpha_i$, with means $\mu_i$, variances $\sigma_i^2$, third central moments $c_{i3}$, and fourth central moments $c_{i4}$. Let $\mathbf{\Gamma} = (\gamma_{ij})$ be an $d \times d$ positive semidefinite matrix, and $\alpha$ be the product measure $\alpha_1 \times \cdots \times \alpha_d$. Inspired by [17], we let $\mathbf{D}$ be the diagonal matrix with $i$th diagonal entry $\sigma_i$ and consider the nonnegative function

$$1 + \frac{1}{2}(\mathbf{x} - \mu)^\top \mathbf{D}^{-1} \mathbf{\Gamma} \mathbf{D}^{-1} (\mathbf{x} - \mu).$$

Its average value is

$$\int \prod_{i=1}^{d} f_i(x_i) \left[ 1 + \frac{1}{2} (\boldsymbol{x} - \boldsymbol{\mu})^\top \boldsymbol{D}^{-1} \boldsymbol{\Gamma} \boldsymbol{D}^{-1} (\boldsymbol{x} - \boldsymbol{\mu}) \right] d\alpha(\boldsymbol{x})$$

$$= 1 + \frac{1}{2} \sum_i \sum_j \mathsf{E} \left[ \frac{(x_i - \mu_i)(x_j - \mu_j)}{\sigma_i \sigma_j} \right] \gamma_{ij}$$

$$= 1 + \frac{1}{2} \sum_i \gamma_{ii}.$$

It follows that the function

$$g(\boldsymbol{x}) = \left[ 1 + \frac{1}{2} \operatorname{tr}(\boldsymbol{\Gamma}) \right]^{-1} \prod_{i=1}^{d} f_i(x_i) \left[ 1 + \frac{1}{2} (\boldsymbol{x} - \boldsymbol{\mu})^\top \boldsymbol{D}^{-1} \boldsymbol{\Gamma} \boldsymbol{D}^{-1} (\boldsymbol{x} - \boldsymbol{\mu}) \right] \tag{1}$$

is a probability density with respect to the measure α. Detailed derivations of Tonda's approximation are found in Sect S1.1 in S1 File. The virtue of the density (1) is that it overcomes the independence restriction and steers the estimated covariance matrix toward the sample covariance matrix of the residuals. Inclusion of the positive semidefinite matrix $\boldsymbol{\Gamma}$ is designed to achieve this goal. For example, if $X_1$ is Gaussian and $X_2$ is Bernoulli, then the density (1) allows one to induce correlation between the two components (one continuous and the other discrete) of the binary random vector $X = (X_1, X_2)^\top$. Note that $g(\boldsymbol{x})$ is technically not a copula because it only approximately preserves the marginal distributions $f_i(x_i)$. Later, we will see how $g(\boldsymbol{x})$ tends to inflate marginal variances and accommodate correlation. This can be a blessing rather than a curse if correlation is present or the true marginal distributions are over-dispersed compared to the assumed marginal distributions.

### 2.3 Moments

Let $\boldsymbol{Y} = (Y_1, \ldots, Y_d)^\top$ be a random vector distributed as $g(\boldsymbol{x})$. To calculate the mean of $Y_k$, note that our independence assumption implies

$$\int (x_k - \mu_k) g(\boldsymbol{x}) \alpha(\boldsymbol{x})$$

$$= \left[ 1 + \frac{1}{2} \operatorname{tr}(\boldsymbol{\Gamma}) \right]^{-1} \frac{1}{2} \sum_i \sum_j \mathsf{E} \left[ (x_k - \mu_k) \frac{(x_i - \mu_i)(x_j - \mu_j)}{\sigma_i \sigma_j} \right] \gamma_{ij}$$

$$= \left[ 1 + \frac{1}{2} \operatorname{tr}(\boldsymbol{\Gamma}) \right]^{-1} \frac{c_{k3} \gamma_{kk}}{2 \sigma_k^2}.$$

Hence, if $\kappa_{k3}$ is the skewness of $X_k$, then

$$\mathsf{E}(Y_k) = \mu_k + \left[ 1 + \frac{1}{2} \operatorname{tr}(\boldsymbol{\Gamma}) \right]^{-1} \frac{c_{k3} \gamma_{kk}}{2 \sigma_k^2}$$

$$= \mu_k + \left[ 1 + \frac{1}{2} \operatorname{tr}(\boldsymbol{\Gamma}) \right]^{-1} \frac{\sigma_k \kappa_{k3} \gamma_{kk}}{2}$$

$$= \mu_k + \frac{\sigma_k \kappa_{k3} \gamma_{kk}}{2} + O(\|\boldsymbol{\Gamma}\|^2)$$

for any matrix norm $\|\boldsymbol{\Gamma}\|$. The mean $\mathsf{E}(Y_k)$ is close to $\mu_k$ when the diagonal entries of $\boldsymbol{\Gamma}$ and, hence $\|\boldsymbol{\Gamma}\|$ itself, are small.
   To calculate the covariance matrix of $\boldsymbol{Y}$, note that

$$\int (x_k - \mu_k)(x_l - \mu_l)g(\boldsymbol{x})d\alpha(\boldsymbol{x})$$

$$= \left[1 + \frac{1}{2}\text{tr}(\boldsymbol{\Gamma})\right]^{-1} 1_{\{k=l\}}\sigma_k^2 + \left[1 + \frac{1}{2}\text{tr}(\boldsymbol{\Gamma})\right]^{-1}$$

$$\times \frac{1}{2}\sum_i \sum_j E\left[(x_k - \mu_k)(x_l - \mu_l)\frac{(x_i - \mu_i)(x_j - \mu_j)}{\sigma_i \sigma_j}\right]\gamma_{ij}.$$

The indicated expectations relative to $\prod_{i=1}^d f_i(x_i)$ reduce to

$$E[(x_k - \mu_k)(x_l - \mu_l)(x_i - \mu_i)(x_j - \mu_j)]$$

$$= \begin{cases} c_{k4} & k = l = i = j \\ \sigma_k^2 \sigma_i^2 & k = l \neq i = j \\ \sigma_k^2 \sigma_l^2 & k = i \neq l = j \\ \sigma_k^2 \sigma_l^2 & k = j \neq l = i \\ 0 & \text{otherwise .} \end{cases}$$

When $k = l$ and $\kappa_{k4}$ is the kurtosis of $X_k$,

$$\int (x_k - \mu_k)^2 g(\boldsymbol{x})d\alpha(\boldsymbol{x})$$

$$= \left[1 + \frac{1}{2}\text{tr}(\boldsymbol{\Gamma})\right]^{-1}\left[\sigma_k^2 + \frac{1}{2}\frac{c_{k4}\gamma_{kk}}{\sigma_k^2} + \frac{1}{2}\sigma_k^2 \sum_{i \neq k} \gamma_{ii}\right]$$

$$= \left[1 + \frac{1}{2}\text{tr}(\boldsymbol{\Gamma})\right]^{-1}\sigma_k^2\left[1 + \frac{\kappa_{k4}\gamma_{kk}}{2} + \frac{1}{2}\sum_{i \neq k} \gamma_{ii}\right]$$

$$+ \frac{\sigma_k^2 \kappa_{k4}\gamma_{kk}}{2} + \frac{\sigma_k^2}{2}\sum_{i \neq k} \gamma_{ii} + O(\|\boldsymbol{\Gamma}\|^2)$$

$$= \sigma_k^2\left[1 + \frac{(\kappa_{k4} - 1)\gamma_{kk}}{2} + \frac{1}{2}\sum_i \gamma_{ii}\right] + O(\|\boldsymbol{\Gamma}\|^2)$$

$$= \sigma_k^2\left[1 + \frac{(\kappa_{k4} - 1)\gamma_{kk}}{2}\right] + O(\|\boldsymbol{\Gamma}\|^2).$$

Because $[E(Y_k - \mu_k)]^2 = O(\|\boldsymbol{\Gamma}\|^2)$, we find that

$$\text{Var}(Y_k) = E[(Y_k - \mu_k)^2] - [E(Y_k - \mu_k)]^2$$

$$= \sigma_k^2\left[1 + \frac{(\kappa_{k4} - 1)\gamma_{kk}}{2}\right] + O(\|\boldsymbol{\Gamma}\|^2).$$

Because the kurtosis $\kappa_{k4} \geq 1$, the multiplier $\kappa_{k4} - 1$ of $\gamma_{kk}$ is nonnegative, and the variance is inflated for $\|\boldsymbol{\Gamma}\|$ small.
When $k \neq l$,

$$\int (x_k - \mu_k)(x_l - \mu_l)g(\boldsymbol{x})d\alpha(\boldsymbol{x}) = \left[1 + \frac{1}{2}\text{tr}(\boldsymbol{\Gamma})\right]^{-1}\frac{1}{2}2\sigma_k\sigma_l\gamma_{kl}$$

$$= \sigma_k\sigma_l\gamma_{kl} + O(\|\boldsymbol{\Gamma}\|^2).$$

Hence, the covariance and correlation satisfy

$$
\begin{aligned}
\mathrm{Cov}(Y_k, Y_l) &= \mathrm{Cov}(Y_k - \mu_k, Y_l - \mu_l) \\
&= \mathrm{E}[(Y_k - \mu_k)(Y_l - \mu_l)] - \mathrm{E}(Y_k - \mu_k)\mathrm{E}(Y_l - \mu_l) \\
&= \sigma_k \sigma_l \gamma_{kl} + O(\|\boldsymbol{\Gamma}\|^2)
\end{aligned}
$$

$$
\mathrm{Corr}(Y_k, Y_l) = \frac{\gamma_{kl}}{\sqrt{1 + \frac{(\kappa_{k4}-1)\gamma_{kk}}{2} + O(\|\boldsymbol{\Gamma}\|^2)}\sqrt{1 + \frac{(\kappa_{l4}-1)\gamma_{ll}}{2} + O(\|\boldsymbol{\Gamma}\|^2)}}.
$$

As a check, the quantities $\mathrm{E}(Y_k)$, $\mathrm{Var}(Y_k)$, and $\mathrm{Cov}(Y_k, Y_l)$ reduce to the correct values $\mu_k$, $\sigma_k^2$, and 0, respectively, when $\boldsymbol{\Gamma} = \boldsymbol{0}$.

## 2.4 Marginal and conditional distributions

Let $S$ be a subset of $\{1, \ldots, d\}$ with complement $T$. To simplify notation, suppose $S = \{1, 2, \ldots, s\}$. Now write

$$
\boldsymbol{Y} = \begin{pmatrix} \boldsymbol{Y}_S \\ \boldsymbol{Y}_T \end{pmatrix}, \quad \boldsymbol{r} = \begin{pmatrix} \boldsymbol{r}_S \\ \boldsymbol{r}_T \end{pmatrix}, \quad \boldsymbol{\Gamma} = \begin{pmatrix} \boldsymbol{\Gamma}_S & \boldsymbol{\Gamma}_{ST} \\ \boldsymbol{\Gamma}_{ST}^\top & \boldsymbol{\Gamma}_T \end{pmatrix}, \quad \alpha = \alpha_S \times \alpha_T,
$$

where $\boldsymbol{r}$ is the vector $\boldsymbol{D}^{-1}(\boldsymbol{Y} - \boldsymbol{\mu})$ of standardized residuals. The marginal density of $\boldsymbol{Y}_S$ is

$$
\begin{aligned}
&\left[1 + \frac{1}{2}\mathrm{tr}(\boldsymbol{\Gamma})\right]^{-1} \prod_{i \in S} f_i(y_i) \int \prod_{i \in T} f_i(y_i)\left[1 + \frac{1}{2}\boldsymbol{r}^\top \boldsymbol{\Gamma} \boldsymbol{r}\right] d\alpha_T(\boldsymbol{y}_T) \\
&= \left[1 + \frac{1}{2}\mathrm{tr}(\boldsymbol{\Gamma})\right]^{-1} \prod_{i \in S} f_i(y_i)\left[1 + \frac{1}{2}\boldsymbol{r}_S^\top \boldsymbol{\Gamma}_S \boldsymbol{r}_S + \frac{1}{2}\mathrm{tr}(\boldsymbol{\Gamma}_T)\right].
\end{aligned}
$$

To derive the conditional density of $\boldsymbol{Y}_S$ given by $\boldsymbol{Y}_T$, we divide the joint density by the marginal density of $\boldsymbol{Y}_T$. This action produces the conditional density

$$
d_S \prod_{i \in S} f_i(y_i)\left[1 + \frac{1}{2}\boldsymbol{r}^\top \boldsymbol{\Gamma} \boldsymbol{r}\right]
$$

with normalizing constant $d_S = \left[1 + \frac{1}{2}\boldsymbol{r}_T^\top \boldsymbol{\Gamma}_T \boldsymbol{r}_T + \frac{1}{2}\mathrm{tr}(\boldsymbol{\Gamma}_S)\right]^{-1}$. From this density, our well-rehearsed arguments lead to the conditional mean

$$
\mathrm{E}(Y_k \mid \boldsymbol{Y}_T) = \mu_k + d_S\left[\frac{c_{k3}\gamma_{kk}}{2\sigma_k^2} + \frac{1}{\sigma_k}\sum_{j \in T} r_j \gamma_{jk}\right] = \mu_k + \frac{c_{k3}\gamma_{kk}}{2\sigma_k^2} + O(\|\boldsymbol{\Gamma}\|^2)
$$

for $k \in S$. The corresponding conditional variance is

$$
\mathrm{Var}(Y_k \mid \boldsymbol{Y}_T) = \sigma_k^2 + \frac{1}{2}\left(\frac{c_{k4}}{\sigma_k^2} - \sigma_k^2\right)\gamma_{kk} + \sum_{j \in T} \frac{c_{k3} r_j \gamma_{kj}}{\sigma_k} + O(\|\boldsymbol{\Gamma}\|^2),
$$

and the corresponding conditional covariances are

$$
\mathrm{Cov}(Y_k, Y_l \mid \boldsymbol{Y}_T) = \sigma_k \sigma_l \gamma_{kl} + O(\|\boldsymbol{\Gamma}\||^2)
$$

for $k \in S$, $l \in S$, and $k \neq l$. It is noteworthy that to order $O(\|\Gamma\|^2)$, the conditional and marginal means agree, and the conditional and marginal covariances agree.

## 2.5 Generation of random deviates

To generate a random vector from the density (1), we first sample $Y_1$ from its marginal density

$$\left[1 + \frac{1}{2}\operatorname{tr}(\Gamma)\right]^{-1} f_1(y_1)\left(1 + \frac{\gamma_{11}}{2}r_1^2 + \frac{1}{2}\sum_{j=2}^{d}\gamma_{jj}\right),$$

and then sample the subsequent components $Y_i$ from their conditional distributions $Y_i \mid Y_1, \ldots, Y_{i-1}, \forall i \in [1, d]$. If we denote the set $\{1, \ldots, i-1\}$ by $[i-1]$, then the conditional density of $Y_i$ given the previous components is

$$d_{[i-1]}^{-1} f_i(y_i)\left[d_{[i-1]} + r_i \sum_{j=1}^{i-1} r_j\gamma_{ij} + \frac{\gamma_{ii}}{2}(r_i^2 - 1)\right],$$

where $d_{[i-1]} = 1 + \frac{1}{2}r_{[i-1]}^{\top}\Gamma_{[i-1]}r_{[i-1]} + \frac{1}{2}\sum_{j=i}^{d}\gamma_{jj}$.

When the densities $f_i(y_i)$ are discrete, each stage of sampling is straightforward. Consider any random variable $Z$ with nonnegative integer values, discrete density $p_i = \operatorname{Pr}(Z = i)$, and mean v. The inverse method of random sampling reduces to a sequence of comparisons. We partition the interval $[0, 1]$ into subintervals with the $i$th subinterval of length $p_i$. To sample $Z$, we draw a uniform random deviate $U$ from $[0, 1]$ and return the deviate $j$ determined by the conditions $\sum_{i=1}^{j-1} p_i \leq U < \sum_{i=1}^{j} p_i$. The process is most efficient when the largest $p_i$ occur first. This suggests that we let $k$ denote the least integer $\lfloor \nu \rfloor$ and rearrange the probabilities in the order $p_k, p_{k+1}, p_{k-1}, p_{k+2}, p_{k-2}, \ldots$ This tactic is apt put most of the probability mass first and render sampling efficient.

When the densities $f_i(y_i)$ are continuous, each stage of sampling is probably best performed by inverse transform sampling. This requires calculating distribution functions and forming their inverses, either analytically or by Newton's method. The required distribution functions assume the form

$$\int_{-\infty}^{x} f(y)[a_0 + a_1(y - \mu) + a_2(y - \mu)^2]\,dy = \int_{-\infty}^{x} f(y)[b_0 + b_1 y + b_2 y^2]\,dy.$$

The integrals $\int_{-\infty}^{x} f(y)y^i\,dy$ are available as special functions for Gaussian, beta, and gamma densities $f(y)$. For instance, if $\phi(y) = \frac{1}{2\pi}e^{-y^2/2}$ is the standard normal density and $\Phi(x)$ is the standard normal distribution, then

$$\int_{-\infty}^{x} y\phi(y)\,dy = -\phi(x) \quad \text{and} \quad \int_{-\infty}^{x} y^2\phi(y)\,dy = \Phi(x) - x\phi(x).$$

To avoid overburdening the text with classical mathematics, we omit further details. Additional derivations can be found in Sect S1.2 in S1 File.

## 2.6 Model for longitudinal data

As a concrete example, we now illustrate how to use the approximate-copula model for the longitudinal setting, with $n$ independent subjects. The response vector $y_i$ for subject $i$ consists of $d_i$ traits values and $p$ covariates (features) $X_i \in \mathbb{R}^{d_i \times p}$ per trait. The component $y_{ij}$ represents the measured trait of subject $i$ at time $j$. The data matrix $X_i$ may include both time-varying features (for example medication use) and time-invariant features (for example gender). Linear mixed

models [5,18] are a sensible modeling choice when the trait values in $\boldsymbol{y}_i$ are continuous. When measurements are discrete, the approximate-copula model often runs orders of magnitude faster than generalized linear mixed models and leads to more plausible loglikelihoods. Sects 3.1 and 3.4 cover both simulated and real longitudinal data.

In the approximate-copula model of longitudinal data, the random vector $Y = (Y_1, ..., Y_{d_i})$ follows the approximate-copula density appearing in Eq (1). Marginal means of the base distribution are linked to covariates through

$$\boldsymbol{\mu}_i(\boldsymbol{\beta}) = g^{[-1]}(\boldsymbol{X}_i\boldsymbol{\beta}), \quad \boldsymbol{\beta} \in \mathbb{R}^p, \tag{2}$$

where the inverse link function $g^{[-1]}(z)$ is applied component-wise. Although it is not strictly necessary, we assume that each of the $d_i$ measurements follows the same base distribution. This simplifying assumption holds, for example, with repeated blood-pressure measurements.

Because each subject $i$ can be measured at a different number $d_i$ of time points, the choice of positive semidefinite matrices $\boldsymbol{\Gamma}_i$ is constrained. We explore three structured options:

$$\boldsymbol{\Gamma}_i = \sum_{j=1}^{m} \theta_j \boldsymbol{\Omega}_{ij}$$

$$\boldsymbol{\Gamma}_i = \sigma^2 \times \begin{bmatrix} 1 & \rho & \rho^2 & \rho^3 & \cdots & \rho^{d_i-1} \\ \rho & 1 & \rho & \rho^2 & \cdots & \\ & & \cdots & & & \\ & & \cdots & \rho & 1 & \rho \\ \rho^{d_i-1} & \rho^{d_i-2} & \cdots & \rho^2 & \rho & 1 \end{bmatrix}$$

$$\boldsymbol{\Gamma}_i = \sigma^2 \times \left[ \rho \mathbf{1}_{d_i} \mathbf{1}_{d_i}^\top + (1-\rho)\boldsymbol{I}_{d_i} \right].$$

$$\boldsymbol{\Gamma}_i = \sigma^2 \times \left[ \rho \mathbf{1}_{d_i} \mathbf{1}_{d_i}^\top + (1-\rho)\boldsymbol{I}_{d_i} \right].$$

These traditional choices define the variance component (VC) model decomposing $\boldsymbol{\Gamma}_i$ into a linear combination of $m$ known positive semidefinite matrices $\boldsymbol{\Omega}_{ij} = (\omega_{ijkl})$ against unknown nonnegative variance components $\boldsymbol{\theta} = (\theta_1, ..., \theta_m)$, the auto-regressive (AR1) model, and the compound symmetric (CS) model. The $\theta_i \geq 0$ must be estimated in the VC model, while $\sigma^2 \geq 0$ and $\rho$ must be estimated in the AR1 and CS models. Sect 2.9.2 describes how estimation is performed. The $\boldsymbol{\Gamma}_i$ parameters supplement the regression coefficients $\boldsymbol{\beta}$ of the base distributions.

## 2.7 Model for unstructured multivariate data

As another example, we will apply the approximate-copula model for analyzing multiple responses. The multivariate model involves $n$ independent samples exhibiting $d$ responses and $p$ covariates. Each component of the approximate-copula response $Y = (Y_1, ..., Y_d)$ is allowed to have a different base distribution $f_j(y)$ and a different inverse link $g_j^{[-1]}(z)$. In contrast to the longitudinal model, we postulate a matrix $\boldsymbol{B} = (\boldsymbol{\beta}_1 \ \dots \ \boldsymbol{\beta}_d)$ of regression coefficients that capture the unique impacts of the $p$ covariates on the $d$ responses. Means are linked to covariates via $\boldsymbol{\mu}_i = g^{[-1]}(\boldsymbol{B}^\top \boldsymbol{x}_i)$. If we define

$$\text{vec}(\boldsymbol{B}) = \begin{bmatrix} \boldsymbol{\beta}_1 \\ \vdots \\ \boldsymbol{\beta}_d \end{bmatrix}_{pd \times 1} \quad \text{and} \quad \boldsymbol{X}_i = \begin{bmatrix} \boldsymbol{x}_i^\top & & 0 \\ & \ddots & \\ 0 & & \boldsymbol{x}_i^\top \end{bmatrix}_{d \times pd},$$

then we can apply the previous notation

$$\mu_i = g^{[-1]}[X_i \, \text{vec}(B)].$$ (3)

This notational change allows us to estimate a parameter vector $\beta \equiv \text{vec}(B)$, with the understanding that $\beta$ has length $p$ for longitudinal data and length $pd$ for multivariate data.

Because the positive semidefinite matrices $\Gamma_i$ are constant across samples, one can estimate an single unstructured matrix

$$\Gamma = LL^\top \quad \in \quad \mathbb{R}^{d \times d},$$

where $L$ is the lower-triangular Cholesky decomposition of $\Gamma$. In the interests of parsimony, one could replace $L$ by a low-rank matrix, but we will not pursue this suggestion further. In summary, the multivariate approximate-copula model is parametrized by $\text{vec}(B) \in \mathbb{R}^{dp}$ mean-effect parameters in addition to the $d(d + 1)/2$ nonzero parameters of the Cholesky decomposition $L = (\ell_{ij})$. The requirement $\ell_{ii} \geq 0$ for all $i$ is the only constraint on the parameter space.

## 2.8 Genome-wide association studies

The ability to analyze correlated non-Gaussian responses is particularly pertinent to genome-wide association studies (GWAS). Most current methods for multivariate GWAS are based on Gaussian approximations [3,23]. The normality assumption makes it challenging to examine multiple traits where some are dichotomous or integer-valued. Here we illustrate how to apply approximate-copula models to GWAS data involving mixtures of continuous, binary, and discrete outcomes. In addition, our approximate-copula framework easily extends to the analysis of non-Gaussian longitudinal traits, such as repeated binary measurements. These traits are common in Electronic Health Record data. For the sake of brevity, we do not pursue this lead here.

In multivariate GWAS, $n$ subjects are measured on $q$ non-genetic covariates (for example sex, age, and diet) and $p$ single-nucleotide polymorphisms (SNPs), where $p \sim 10^6$. Together the covariates influence a small set of $d$ traits (phenotypes). Due to the high-dimensionality of the problem, typically each SNP is examined separately. If $\alpha \in \mathbb{R}^d$ represents the mean effect of a SNP on each of $d$ phenotypes, then the null hypothesis

$$H_0 : \alpha_1 = ... = \alpha_d = 0$$ (4)

is pertinent. To test the hypothesis (4), a likelihood ratio tests (LRT) is certainly possible. This involves maximizing the loglikelihood under the null hypothesis $H_0$ and comparing it to the likelihood under the alternative hypothesis $H_a$ where $\alpha$ varies. If $\mathcal{L}_0$ and $\mathcal{L}_a$ denote the respective maximum loglikelihoods, then the likelihood ratio statistic follows the distribution

$$2(\mathcal{L}_a - \mathcal{L}_0) \sim \chi_d^2.$$ (5)

SNPs can be selected by applying the Bonferroni's correction to the resulting p-values. For GWAS with human subjects, the stringent cutoff p-value $5 \times 10^{-8}$ is standard.

Although this strategy is conceptually simple, it requires fitting $p \sim 10^6$ alternative models. To alleviate the computational burden, we calculate the LRT on only the most promising SNPs. One can screen for the top SNPs by examining the gradient $\nabla_\alpha \mathcal{L} \in \mathbb{R}^d$ under the maximum likelihood estimates of the null model estimates $(\hat{\beta}, \hat{L})$. The $\ell_1$ norm $\|\nabla_\alpha \mathcal{L}\|_1$ quantifies the signal strength of the SNP under consideration. We will derive the gradient $\nabla_\alpha \mathcal{L}$ later. Computing the gradient under the null is much faster than fitting a full likelihood model under the alternative. We show in Sect S1.6.1 in S1 File that ordering SNPs by $\|\nabla_\alpha \mathcal{L}\|_1$ is strongly correlated with ordering them by $-\log_{10}(\text{p-value})$. Algorithm 1 summarizes

our fast multivariate GWAS procedure. Table N in S1 File displays a QQ plot for Algorithm 1, showing that the resulting p-values are valid.

### Algorithm 1. Fast likelihood-ratio tests for multivariate GWAS.

```
Data: phenotypes Y ∈ ℝⁿˣᵈ, covariates X ∈ ℝⁿˣq, genotypes G ∈ ℝⁿˣp
1  ℒ₀ ← fit(Y, X)        ## maximum loglikelihood under H₀
2  for j = 1, 2, ..., p do
3  |   rⱼ ← ‖∇_α ℒ‖₁      ## normed gradient of SNP j under H₀
4  end
5  Sort from largest to smallest (r[1], ..., r[p]) ← sort(r₁, ..., rₚ)
6  for j = 1, 2, ..., p do
7  |   gⱼ ← jth SNP genotypes corresponding to r[j]
8  |   ℒ_a ← fit(Y, X, gⱼ)    ## maximum loglikelihood under H_a
9  |   LRT: 2(ℒ_a − ℒ₀) ∼ χ²_d
10 |   terminate: if the p-value is larger than a preset level
11 end
```

## 2.9 Parameter estimation

Throughout this section, the vector $\beta$ denotes the mean effects tied to the base distributions through Eqs (2) and (3). The vector $\theta$ summarizes the dependency parameters determining the positive semidefinite matrices $\Gamma_i$. For example, in the longitudinal AR1 model, $\theta = (\sigma^2, \rho)$, and in the multivariate model, $\theta = \text{vech}(L)$, where $\text{vech}(L)$ captures the lower triangle of the Cholesky decomposition $L$.

**2.9.1 Mean components.** Consider $n$ independent realizations $y_i$ from the approximate-copula density (1). Each of these may be of a different dimension $d_i$ and possess a different mean vector $\mu_i(\beta)$, positive semidefinite matrice $\Gamma_i(\theta) = [\gamma_{ijk}(\theta)]$, and component densities $f_{ij}(y_{ij} \mid \beta)$. If $r_i(\beta)$ denotes the vector $D_i^{-1}(y_i - \mu_i)$ of standardized residuals for sampling unit $i$, then the loglikelihood of the sample is

$$\mathcal{L}(\beta, \theta) = -\sum_{i=1}^{n} \ln \left\{ 1 + \frac{1}{2}\text{tr}[\Gamma_i(\theta)] \right\} + \sum_{i=1}^{n}\sum_{j=1}^{d_i} \ln f_{ij}(y_{ij} \mid \beta)$$

$$+ \sum_{i=1}^{n} \ln \left[ 1 + \frac{1}{2}r_i(\beta)^\top \Gamma_i(\theta) r_i(\beta) \right].$$

The score (gradient of the loglikelihood) with respect to $\beta$ is clearly

$$\nabla_\beta \mathcal{L}(\beta, \theta) = \sum_{i=1}^{n}\sum_{j=1}^{d_i} \nabla \ln f_{ij}(y_{ij} \mid \beta) + \sum_{i=1}^{n} \frac{\nabla r_i(\beta)^\top \Gamma_i(\theta) r_i(\beta)}{1 + \frac{1}{2}r_i(\beta)^\top \Gamma_i(\theta) r_i(\beta)},$$

where $\nabla r_i(\beta)^\top = dr_i(\beta)$ is the differential (Jacobi matrix) of the vector $r_i(\beta)$. An easy calculation shows that $\nabla r_i(\beta)$ has entries

$$\nabla r_{ij}(\beta) = -\frac{1}{\sigma_{ij}(\beta)}\nabla \mu_{ij}(\beta) - \frac{1}{2}\frac{y_{ij} - \mu_{ij}(\beta)}{\sigma_{ij}^3(\beta)}\nabla \sigma_{ij}^2(\beta).$$

The Hessian (second differential) of the loglikelihood with respect to $\beta$ is

$$d^2_{\beta}\mathcal{L}(\beta, \theta) = \sum_{i=1}^{n}\sum_{j=1}^{d_i} d^2 \ln f_{ij}(y_{ij} \mid \beta) - \sum_{i=1}^{n} \frac{[\nabla r_i(\beta)\Gamma_i r_i(\beta)][\nabla r_i(\beta)\Gamma_i r_i(\beta)]^{\top}}{\left[1 + \frac{1}{2}r_i(\beta)^{\top}\Gamma_i r_i(\beta)\right]^2}$$

$$+ \sum_{i=1}^{n} \frac{\nabla r_i(\beta)\Gamma_i dr_i(\beta)}{1 + \frac{1}{2}r_i(\beta)^{\top}\Gamma_i r_i(\beta)} + \sum_{i=1}^{n}\sum_{j=1}^{d_i} \frac{e_j^{\top}\Gamma_i r_i(\beta)d^2 r_{ij}(\beta)}{1 + \frac{1}{2}r_i(\beta)\Gamma_i r_i(\beta)},$$

where

$$d^2 r_{ij}(\beta) = -\frac{1}{\sigma_{ij}(\beta)}d^2\mu_{ij}(\beta) + \frac{1}{2}\frac{1}{\sigma_{ij}^3(\beta)}\nabla\sigma_{ij}^2(\beta)d\mu_{ij}(\beta) - \frac{1}{2}\frac{y_{ij} - \mu_{ij}(\beta)}{\sigma_{ij}^3(\beta)}d^2\sigma_{ij}^2(\beta)$$

$$+ \frac{1}{2}\left[\frac{1}{\sigma_{ij}^3(\beta)}\nabla\mu_{ij}(\beta) + \frac{3}{2}\frac{y_{ij} - \mu_{ij}(\beta)}{\sigma_{ij}^5(\beta)}\nabla\sigma_{ij}^2(\beta)\right]d\sigma_{ij}^2(\beta).$$

Calculation of $\nabla r_{ij}$ and $d^2 r_{ij}$ requires computing $\nabla\mu_{ij}(\beta)$, $d^2\mu_{ij}(\beta)$, $\nabla\sigma_{ij}^2(\beta)$, and $d^2\sigma_{ij}^2(\beta)$ by repeated application of the chain rule. A full example appear in Sect S1.3.6 in S1 File.

In searching the likelihood surface, it is often beneficial to approximate the observed information by a positive semidefinite matrix. This suggests replacing the observed information matrix $-d^2 \ln f_{ij}(y_{ij} \mid \beta)$ by the expected information matrix $J_{ij}(\beta)$ under the base model and dropping indefinite matrices in the exact Hessian. These steps give the approximate Hessian

$$d^2_{\beta}\mathcal{L} \approx -\sum_{i=1}^{n}\sum_{j=1}^{d_i} J_{ij}(\beta) - \sum_{i=1}^{n} \frac{[\nabla r_i(\beta)\Gamma_i(\theta)r_i(\beta)][\nabla r_i(\beta)\Gamma_i(\theta)r_i(\beta)]^{\top}}{\left[1 + \frac{1}{2}r_i(\beta)^{\top}\Gamma_i r_i(\beta)\right]^2},$$

which is clearly negative semidefinite. These formulas provide the ingredients for implementing Newton's method or a quasi-Newton method for updating $\beta$.

**2.9.2 Derivatives pertinent to VC models.** Maximization of the loglikelihood depends on the derivatives of positive semidefinite matrices $\Gamma_i(\theta)$ through the functions $\text{tr}(\Gamma_i)$ and $r_i(\beta)^{\top}\Gamma_i r_i(\beta)$. Here we present details of the longitudinal VC model and relegate others to Sect S1.3.3 in S1 File. Recall that the longitudinal VC model involves the decomposition

$$\Gamma_i(\theta) = \sum_{j=1}^{m} \theta_j \Omega_{ij}$$

of $\Gamma_i$ into a linear combination of $m$ known positive semidefinite matrices $\Omega_{ij} = (\omega_{ijkl})$ against unknown nonnegative variance components $\theta = (\theta_1, ..., \theta_m)$. Assuming there are no shared mean and $\Gamma_i$ parameters,

$$\text{tr}(\Gamma_i) = \sum_{j=1}^{m} \theta_j \text{tr}(\Omega_{ij}) \quad = \quad \theta^{\top} c_i$$

$$r_i(\beta)^{\top}\Gamma_i r_i(\beta) = \sum_{j=1}^{m} \theta_j r_i(\beta)^{\top}\Omega_{ij}r_i(\beta) \quad = \quad \theta^{\top} b_i,$$

in obvious notation. The part of the loglikelihood relevant to estimation of $\theta$ can be expressed as

$$\mathcal{L}(\theta) = \sum_{i=1}^{n} \ln(1 + \theta^{\top} b_i) - \sum_{i=1}^{n} \ln(1 + \theta^{\top} c_i).$$

Consequently, the score and Hessian with respect to $\theta$ are

$$\nabla_{\theta} \mathcal{L}(\beta, \theta) = \sum_{i=1}^{n} \frac{b_i}{1 + \theta^{\top} b_i} - \sum_{i=1}^{n} \frac{c_i}{1 + \theta^{\top} c_i}$$

$$d_{\theta}^2 \mathcal{L}(\beta, \theta) = - \sum_{i=1}^{n} \frac{b_i b_i^{\top}}{(1 + \theta^{\top} b_i)^2} + \sum_{i=1}^{n} \frac{c_i c_i^{\top}}{(1 + \theta^{\top} c_i)^2}.$$

**2.9.3 Derivatives of an unstructured Gamma matrix.** If we parametrize $\Gamma$ using the Cholesky factor $\Gamma = LL^{\top}$ as suggested in Sect 2.7, then the loglikelihood function can be written as

$$\mathcal{L}(\beta, L) = - \sum_{i=1}^{n} \ln \left[ 1 + \frac{1}{2} \mathrm{tr}(LL^{\top}) \right] + \sum_{i=1}^{n} \sum_{j=1}^{d} \ln f_{ij}(y_{ij} \mid \beta, \gamma)$$

$$+ \sum_{i=1}^{n} \ln \left[ 1 + \frac{1}{2} r_i(\beta)^{\top} LL^{\top} r_i(\beta) \right].$$

The directional derivatives

$$d_V \frac{1}{2} \mathrm{tr}(LL^{\top}) = \mathrm{tr}(LV^{\top})$$

$$d_V \frac{1}{2} r_i(\beta)^{\top} LL^{\top} r_i(\beta) = r_i(\beta)^{\top} LV^{\top} r_i(\beta)$$

follow from the standard rules of differentiation [7,11]. For coding purposes it is easier to invoke reverse-mode automatic differentiation tools rather than implementing the Hessian in closed form [12]. Indeed, if we write

$$\gamma \equiv \begin{bmatrix} \mathrm{vec}(B) \\ \mathrm{vech}(L) \end{bmatrix} = \begin{bmatrix} \beta \\ \theta \end{bmatrix} \in \mathbb{R}^{pd+d(d+1)/2},$$

then the loglikelihood can be viewed as a vector-input scalar-output function $\mathcal{L}(\gamma)$. At some sacrifice of computational speed, automatic differentiation will evaluate $\nabla \mathcal{L}(\gamma)$ and $d^2 \mathcal{L}(\gamma)$ at a current parameter vector $\gamma$. These derivatives enable implementation of Newton's method or a quasi-Newton method for fitting the multivariate approximate-copula model.

**2.9.4 Nuisance parameter estimation.** Many distributions are parametrized by additional nuisance parameters that also require estimation. In general, the strategy for estimating these is similar to the strategy for estimating the mean or variance parameters. For brevity, we illustrate this procedure concretely for the Gaussian and negative binomial distributions in S1 File Sects S1.3.4 and S1.4.

**2.9.5 Initialization.** Most optimization algorithms benefit from good starting values. The obvious candidate for $\beta$ is the maximum likelihood estimate delivered by the base model. For structured $\Gamma_i$ models, we use an MM algorithm [6] to initialize variance components. This process is described in Sect S1.3.2 in S1 File. Under the CS and AR1 models, we initialize the variance component $\sigma^2$ by the crude estimate from the MM algorithm treating $\rho = 0$. For unstructured $\Gamma_i$ models, we initialize $L$ by the Cholesky decomposition of sample correlation matrix of $Y$.

## 3 Results

### 3.1 Simulation studies under the longitudinal model

To assess estimation accuracy of the approximate-copula model, we first present simulation studies for the Poisson and negative binomial base distributions with log link function, under the VC parameterization of $\Gamma_i$. Additional simulation studies with different base distributions under the AR(1), CS and VC parameterizations of $\Gamma_i$ are included in Sect S1.5 in S1 File.

In each simulation scenario, the non-intercept entries of the predictor matrix $X_i$ are independent standard normal deviates. True regression coefficients $\beta_{\text{true}} \sim \text{Uniform}(-0.2, 0.2)$. For the negative binomial base, all dispersion parameters are $r_{\text{true}} = 10$. Each simulation scenario was run on 100 replicates for each sample size $n \in \{100, 1000, 10000\}$ and number of observations $d_i \in \{2, 5, 10, 15, 20, 25\}$ per independent sampling unit.

Under the VC parameterization of $\Gamma_i$, the choice $\Gamma_{i,\text{true}} = \theta_{\text{true}} \times \mathbf{1}_{d_i}\mathbf{1}_{d_i}^\top$ allows us to compare to the random intercept GLMM fit using MixedModels.jl. When the random effect term is a scalar, MixedModels.jl uses Gaussian quadrature for parameter estimation. We compare estimates and run-times to the random intercept GLMM fit of MixedModels.jl with 25 Gaussian quadrature points. We conduct simulation studies under two scenarios (simulation I and II). In simulation I, it is assumed that the data are generated by the approximate-copula model with $\theta_{\text{true}} = 0.1$, and in simulation II, it is assumed that the true distribution is the random intercept GLMM with $\theta_{\text{true}} = 0.05$.

**Simulation I:** In this scenario, we simulate datasets under the approximate-copula model as outlined in Sect 5 and compare MLE fits under the approximate-copula model and GLMM. Top panel of Fig 1 helps us assess estimation accuracy and how well the GLMM density approximates the approximate-copula density.

As anticipated, the MSE's across all base distributions decrease as sample size increases. For data simulated under the approximate-copula model, approximate-copula mean squared errors (MSE) are generally lower than GLMM MSE's. GLMM estimated variance components are often zero and stay relatively constant across sample sizes. This confirms the fact that the two models are different in how they handle random effects, particularly with larger sampling units ($d_i > 2$).

**Simulation II:** In the second simulation, we generate datasets under the random intercept Poisson GLMM and compare MLE fits delivered by the two models. Bottom panel of Fig 1 now shed light on how well the approximate-copula density approximates the GLMM density under different magnitudes of the variance components.

As expected, MSE's under GLMM analysis are now generally lower than those under approximate-copula analysis. For the Poisson and negative binomial base distributions with $\theta_{\text{true}} = 0.05$, the bottom panel of Fig 1 indicates biases for the approximate-copula estimates of $(\beta, \theta)$ for larger sampling units ($d_i > 2$) up to sample size $n = 10,000$, similar to the bias observed for $\hat{\theta}$ in the Poisson GLMM example on top panel.

### 3.2 Run times

Run times under simulation I and II are comparable. Table 1 presents average run times and their standard errors in seconds for 100 replicates under simulation II with $\theta_{\text{true}} = 0.01$. All computer runs were performed on a standard 2.3 GHz Intel i9 CPU with 8 cores. Runtimes for the approximate-copula model are presented given multi-threading across 8 cores. We note the current version of MixedModels.jl does not allow for multi-threading across multiple cores.

Because the approximate-copula loglikelihoods contain no determinants or matrix inverses, our software experiences less pronounced increases in computation time as sample and sampling unit sizes grow compared to GLMM implemented in MixedModels.jl. Run times for the approximate-copula model are faster than those of MixedModels.jl for discrete outcomes (Tables 1 and E in S1 File) and slower for Gaussian distributed outcomes (Table F S1 File). This general trend also holds on a per core basis. This discrepancy is hardly surprising since MixedModels.jl takes into account the low-rank structure of $\Omega_i$ in the random intercept linear mixed model (LMM). This tactic reduces the computational complexity per sample from $O(d_i^3)$ to $O(d_i^2)$. More detailed comparisons appear in Sect S1.5.1 in S1 File.

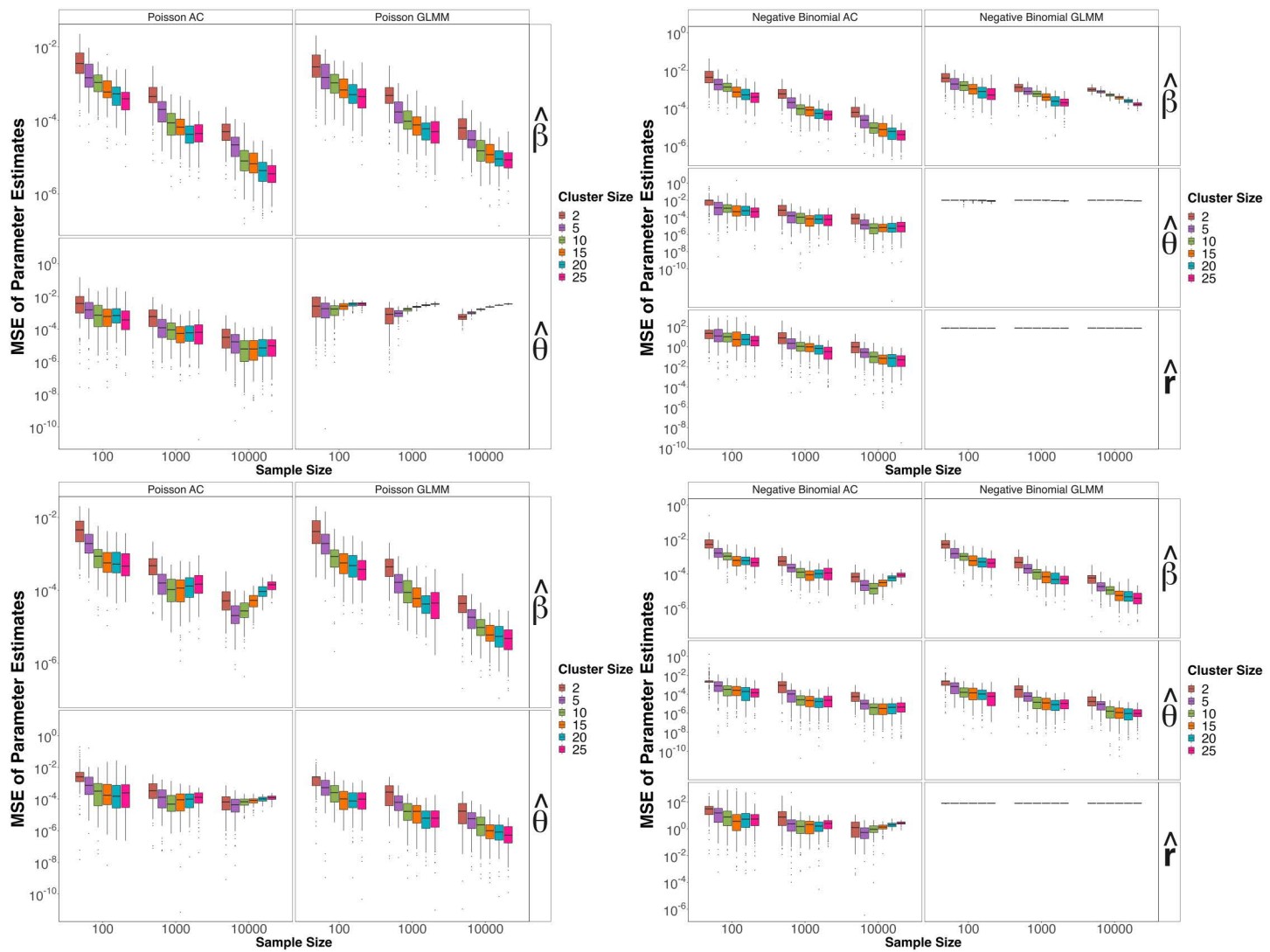

**Fig 1. Simulation study under the longitudinal model.** Top panel features MSE for $\beta$ and $\theta$ under Simulation I setting with Poisson base (left) and Negative binomial base (right). The bottom panel features MSE of $\beta$ and $\theta$ under Simulation II setting with Poisson base (left) and Negative binomial base (right). Here cluster size refers to $d_i$, the number of observations per sample. AC abbreviates approximate-copula and GLMM abbreviates generalized linear mixed model.

Finally, we study the negative binomial base distribution for longitudinal model in more depth. We compared our negative binomial fits with those delivered by the three popular R packages for GLMM estimation in Sect S1.9 in S1 File. Within Julia, MixedModels.jl explicitly warns the user against fitting GLMM's with unknown dispersion parameter $r$. Our software updates $r$ iteratively by Newton's method, holding the other parameters $(\beta, \theta)$ fixed, see Sect S1.3.4 in S1 File for more estimation details.

## 3.3 GWAS simulations

Simulations under the multivariate approximate-copula model demonstrate the potential of Algorithm 1 in GWAS. Specifically, we simulated $n = 5000$ subjects, $p = 15$ covariates, $q = 1000$ independent SNPs, and $d = 4$ correlated responses.

**Table 1. Run times and (standard error of run times) in seconds based on 100 replicates under simulation II with Poisson and negative binomial (NB) Base, $\theta_{\text{true}}$ = 0.01, sampling unit size $d_i$ and sample size $n$. AC abbreviates approximate-copula and GLMM abbreviates generalized linear mixed model.**

| n | $d_i$ | Poisson AC time | Poisson GLMM time | NB AC time | NB GLMM time |
|---|---|---|---|---|---|
| 100 | 2 | 0.021 (<0.001) | 0.022 (0.003) | 0.125 (0.008) | 0.037 (0.003) |
| 100 | 5 | 0.020 (<0.001) | 0.045 (0.003) | 0.095 (0.005) | 0.068 (0.004) |
| 100 | 10 | 0.023 (0.001) | 0.080 (0.004) | 0.105 (0.004) | 0.187 (0.011) |
| 100 | 15 | 0.024 (0.001) | 0.148 (0.006) | 0.105 (0.004) | 0.282 (0.017) |
| 100 | 20 | 0.025 (0.001) | 0.186 (0.007) | 0.112 (0.002) | 0.394 (0.017) |
| 100 | 25 | 0.026 (<0.001) | 0.265 (0.009) | 0.119 (0.003) | 0.461 (0.019) |
| 1000 | 2 | 0.025 (<0.001) | 0.192 (0.007) | 0.163 (0.009) | 0.365 (0.013) |
| 1000 | 5 | 0.030 (<0.001) | 0.516 (0.016) | 0.167 (0.004) | 0.857 (0.033) |
| 1000 | 10 | 0.035 (0.001) | 1.011 (0.022) | 0.243 (0.003) | 1.972 (0.050) |
| 1000 | 15 | 0.040 (<0.001) | 1.402 (0.030) | 0.303 (0.002) | 2.854 (0.064) |
| 1000 | 20 | 0.042 (<0.001) | 1.887 (0.036) | 0.371 (0.002) | 3.722 (0.077) |
| 1000 | 25 | 0.051 (0.001) | 2.531 (0.046) | 0.435 (0.002) | 4.815 (0.089) |
| 10000 | 2 | 0.128 (0.001) | 1.896 (0.032) | 1.169 (0.040) | 3.902 (0.079) |
| 10000 | 5 | 0.154 (0.001) | 4.333 (0.075) | 1.375 (0.020) | 8.598 (0.140) |
| 10000 | 10 | 0.232 (0.002) | 9.545 (0.143) | 2.154 (0.007) | 20.499 (0.303) |
| 10000 | 15 | 0.272 (0.002) | 14.844 (0.249) | 2.78 (0.007) | 29.003 (0.465) |
| 10000 | 20 | 0.336 (0.002) | 21.423 (0.356) | 3.314 (0.007) | 42.952 (0.679) |
| 10000 | 25 | 0.429 (0.003) | 29.324 (0.528) | 4.111 (0.011) | 54.676 (0.861) |

The simulated covariates $\boldsymbol{X} \in \mathbb{R}^{n \times p}$ have entries drawn from $N(0, 1)$. A column of 1's is appended to $\boldsymbol{X}$ to accommodate an intercept. The true effect sizes are randomly sampled from a $\text{Uniform}(0, 0.5)$ distribution. The number of minor alleles for each SNP follows a $\text{Binomial}(2, \rho)$ distribution with $\rho = 0.3$. For simplicity, the true $\boldsymbol{\Gamma}$ is simulated under the AR1 model

$$\boldsymbol{\Gamma} = \begin{pmatrix} 1 & 0.5 & 0.25 & 0.125 \\ 0.5 & 1 & 0.5 & 0.25 \\ 0.25 & 0.5 & 1 & 0.5 \\ 0.125 & 0.25 & 0.5 & 1 \end{pmatrix}.$$

Under this setup we explore 3 simulation scenarios:

**Simulation III:** Here we assume each response $\boldsymbol{y}_i \in \mathbb{R}^d$ is an independent sample from the approximate-copula model. Each of the $d$ components of $\boldsymbol{y}_i$ is randomly chosen from the Gaussian, Bernoulli, and Poisson base distributions. The variance of the Gaussian base is set at $\sigma^2 = 0.5$.

**Simulation IV:** Here the responses $\boldsymbol{y}_i$ are also generated from the approximate-copula model, but now all $d$ components $y_{ij}$ have a Bernoulli base. This scenario is appropriate given multiple correlated case-control responses.

**Simulation V:** Here the responses follow a multivariate Gaussian $\boldsymbol{y}_i = N(\boldsymbol{B}^\top \boldsymbol{x}_i, \boldsymbol{\Gamma})$ distribution. This choice allows us to assess whether approximate-copula fitting correctly collapses to the underlying base model.

To assess power, we randomly choose $k = 10$ causal SNPs. If $\boldsymbol{\alpha} \in \mathbb{R}^d$ denotes the effect of a causal SNP on the $d$ phenotypes, then we impose the constraint $\sum_{i=1}^{d} \alpha_i = s$, where $s$ varies between 0 and 1. Thus, a causal SNP influences each of the $d$ responses, but the magnitudes of its effect are both correlated and random.

Over 100 replicates, we compare the power of Algorithm (2.8) against a penalized regression (IHT) algorithm [3] and the multivariate linear mixed model [23]. As shown in Fig 2, we achieve better power than both IHT and GEMMA when

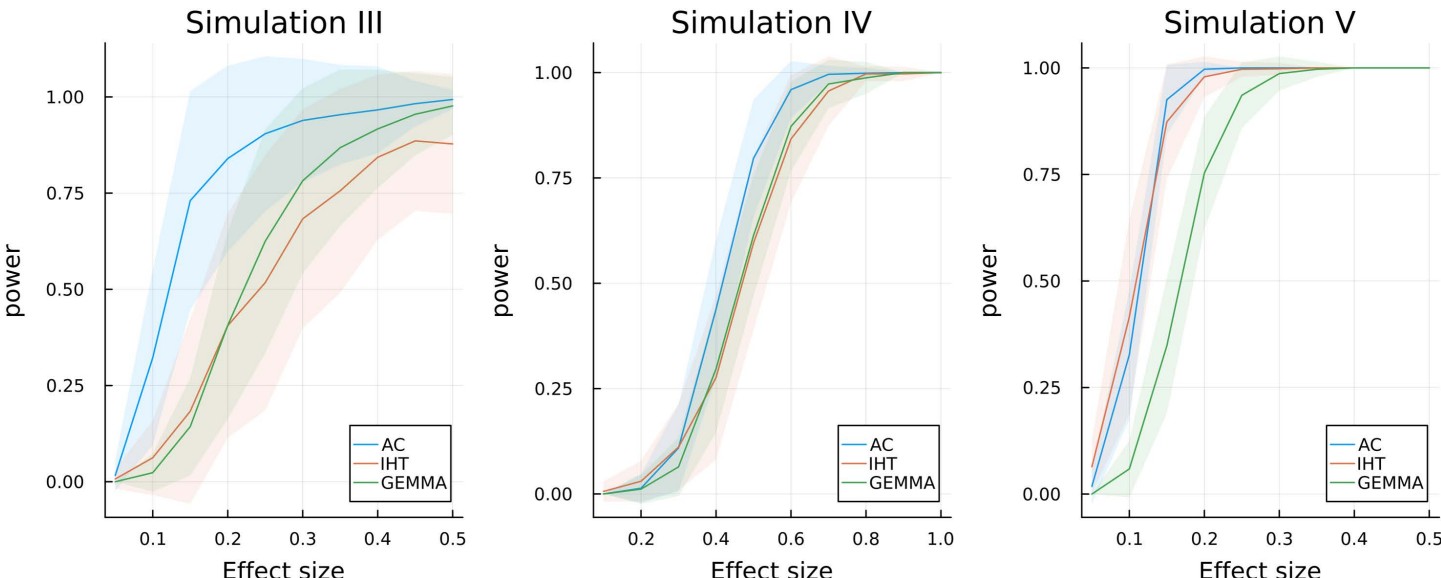

**Fig 2. Power simulation for the proposed multivariate GWAS routine in Algorithm (2.8).** Here AC denotes approximate-copula, IHT denotes iterative hard thresholding, a penalized sparse regression method [3], and GEMMA implements a multivariate linear-mixed model [23]. The colored band represent $\pm 1$ standard deviation.

the data generative process follows the approximate-copula model (simulation III and IV). When the responses are purely Gaussian (simulation V), approximate-copula fitting offers comparable power to IHT. Finally, Sect S1.6.2 in S1 File demonstrates that the approximate-copula model produces valid p-values.

### 3.4 Longitudinal analysis of the NHANES data

For many repeated measurement problems, a simple random intercept model is sufficient to account for correlations between different responses on the same subject. To illustrate this point and the performance of the approximate-copula model, we now turn to a bivariate example from the NHANES I Epidemiologic Followup Study (NHEFS) dataset [4]. In this example, we group the data by subject ID and jointly model the number of cigarettes smoked per day in 1971 and the number of cigarettes smoked per day in 1982 as a bivariate outcome. For fixed effects, we include an intercept and control for sex, age in 1971, and the average price of tobacco in the state of residence. The average price of tobacco is a time-dependent covariate that is adjusted for inflation using the 2008 U.S. consumer price index (CPI). Participants with missing responses or predictors were excluded from the model cohort. A total of $n$ = 1537 NHANES I participants constitute the cohort.

Table 2 compares loglikelihoods and run times under the longitudinal regression model with Poisson, negative binomial, and Bernoulli base distributions as computed by our approximate-copula software, the GLMM package `MixedModels.jl` (a more efficient version of the `lme4` package [1]), and the GEE package `EstimatingEquationsRegression.jl`. For the Bernoulli base distribution, we transformed each count outcome to a binary indicator with value 1 if the number of cigarettes smoked per day is greater than the sample average and value 0 otherwise. The maximum loglikelihood of the approximate-copula model is lower than that of GLMM for the Poisson base and higher than that of GLMM for the negative binomial and Bernoulli bases. The approximate-copula model generally runs faster than GLMM but slightly slower than GEE.

**Table 2. The loglikelihood and run times for the longitudinal NHANES data under the approximate-copula (AC) model, GLMM, and GEE, using three different base distributions. Best values appear in bold faced type. Note the loglikelihoods for GEE are NA since it is not likelihood-based. All $n$ = 1537 sampling units are of size $d_i$ = 2.**

|  | Loglikelihood | Time (seconds) |
|---|---|---|
| AC Poisson | -20690.8 | 0.181 |
| GLMM Poisson | **-15499.5** | 0.529 |
| GEE Poisson | NA | **0.021** |
| AC NB | **-12037.6** | 0.034 |
| GLMM NB | -12047.5 | 0.867 |
| GEE NB | NA | **0.024** |
| AC Bernoulli | **-1938.7** | 0.06 |
| GLMM Bernoulli | -1980.9 | 0.821 |
| GEE Bernoulli | NA | **0.016** |

Table 3 presents detailed parameter estimates for the negative binomial base. Because overdispersion is a feature of this dataset, the Poisson base distribution represents a case of model misspecification as documented in Table G S1 File. The negative binomial base distribution is a better choice for analysis. Under the Poisson base, the approximate-copula model inflates the variance component to account for the over-dispersion. Under the negative binomial base distribution, both our model and `MixedModels.jl` estimate the variance component to be 0. This suggests that no additional overdispersion exists in the data. The estimates for $\beta$ under the approximate-copula model with Poisson base are closer to the more realistic estimates under the negative binomial base than those of GLMM. Sex and age variables in the approximate copula models have larger estimated standard errors than in GLMM or GEE. As a consequence, the predictor age is no longer statistically significant. Given its small estimated effect size, this interpretation is arguably preferable.

### 3.5 Multivariate GWAS on UK biobank data

We also conducted a 3-trait analysis of hypertension related phenotypes from the second release of the UK-Biobank [16]. The underlying traits, average systolic blood pressure (SBP), average diastolic blood pressure (DBP), and body mass index (BMI), are correlated, heritable, and the subject of previous association studies [3]. Although the traits are continuous, we dichotomize both SBP and DBP to illustrate a multivariate analysis with correlated non-continuous responses. Following the clinical definition of stage 2 hypertension [19], we set SBP to 1 if a patient's average SBP is $\geq$ 140 mm Hg and DBP to 1 if a patient's average DBP is $\geq$ 90 mm Hg. Otherwise, these traits are set to 0.

After quality control (see Sect S1.8 in S1 File), the data includes $p$ = 470,228 autosomal SNPs and a subset of $n$ = 80,000 subjects without missing phenotypes. We split this subset of data into 483 contiguous blocks, each containing roughly contiguous 1000 SNPs, and ran Algorithm (2.8) in parallel across them. Each job finished in less than a day. Altogether 617 SNPs pass our threshold for likelihood ratio testing. Fig 3 depicts our results in a GWAS Manhattan plot [20]. Sect S1.10 in S1 File compares our multivariate GWAS result to three separate univariate GWASes interpreted via a Cauchy combination test [9].

After pruning secondary but significant SNPs within 1Mb windows, we uncover roughly 24 association hotspots. The strongest signals come from previously known associations with BMI, such as rs1421085 on chromosome 16, rs10871777 on chromosome 18, and rs13107325 on chromosome 4. These SNPs are known to be associated with BMI independently of SBP and DBP [3]. Because dichotomizing a trait loses information, we expect most discoveries to be associated with BMI. However, we were able to discover SNPs such as rs17367504 on chromosome 1, rs2681492 on chromosome 12, and rs653178 on chromosome 12 previously known to be associated with SBP and DBP independently of BMI.

**Table 3. Comparisons of parameter estimates on the NHANES data under negative binomial approximate-copula (AC) model, GLMM, and GEE. All *n* = 1537 sampling units are of size $d_i$ = 2. Here *r* is the nuisance parameter from negative binomial regression and θ is the variance component.**

| | Estimate | Std. Error | z | Pr(> |z|) | Lower 95% | Upper 95% |
|---|---|---|---|---|---|---|
| **AC** | | | | | | |
| intercept | 2.580 | 0.098 | 26.22 | <1e-99 | 2.387 | 2.773 |
| sex | -0.187 | 0.067 | -2.804 | 0.005 | -0.318 | -0.056 |
| age | -0.009 | 0.009 | -0.961 | 0.336 | -0.027 | 0.009 |
| price | 0.402 | 0.094 | 4.275 | <1.91e-5 | 0.217 | 0.586 |
| *r* | 1.14 | 0.137 | 8.35 | <1e-99 | 0.873 | 1.408 |
| θ | ≈ 0 | 0.052 | -1.92e-7 | 1.0 | -0.102 | 0.102 |
| **GLMM** | | | | | | |
| intercept | 2.580 | 0.117 | 21.92 | <1e-99 | 2.350 | 2.811 |
| sex | -0.187 | 0.027 | -7.04 | <1e-11 | -0.239 | -0.135 |
| age | -0.009 | 0.001 | -7.99 | <1e-14 | -0.011 | -0.007 |
| price | 0.402 | 0.053 | 7.56 | <1e-13 | 0.298 | 0.506 |
| *r* | 1.395 | NA | NA | NA | NA | NA |
| θ | ≈ 0 | NA | NA | NA | NA | NA |
| **GEE** | | | | | | |
| intercept | 2.374 | 0.110 | 21.64 | <1e-99 | 2.16 | 2.59 |
| sex | -0.186 | 0.032 | -5.89 | <1e-8 | -0.248 | -0.124 |
| age | -0.009 | 0.001 | -7.01 | <1e-11 | -0.011 | -0.006 |
| price | 0.507 | 0.045 | 11.14 | <1e-28 | 0.417 | 0.596 |
| *r* | NA | NA | NA | NA | NA | NA |
| θ | NA | NA | NA | NA | NA | NA |

Interestingly, five SNPs rs4500930, rs7721099, rs2293579, rs11191548, and rs34783010 missed in our previous analysis are known to be associated with BMI [10]. It is possible our previous analysis discovered nearby proxies instead. While our current analysis is hardly definitive, it demonstrates that multi-trait GWAS is indeed possible taking proper account of inter-trait correlations.

## 4 Discussion

We propose a new model for analyzing multivariate data based on Tonda's Gaussian copula approximation. Our approximate-copula model enables the analysis of correlated responses and handles random effects needed in applications such as panel and repeated measures data. The approximate-copula model trades Tonda's awkward parameter space constraint for a simple normalizing constant. This allows one to engage in full likelihood analysis under a tractable probability density function with no implicit integrations or matrix inverses. The approximate-copula model is relatively easy to fit and friendly to likelihood ratio hypothesis testing.

For maximum likelihood estimation, we recommend a combination of two numerical methods. The first is a block ascent algorithm that alternates between updating the mean parameters $\beta$ by a version of Newton's method and updating the variance parameters by a minorization-maximization (MM) algorithm. The second method jointly updates $\beta$ and the variances components by a standard quasi-Newton algorithm. The MM algorithm converges quickly to a neighborhood of the MLE but then slows. In contrast, the quasi-Newton struggles at first and then converges quickly. Thus, we start with the block ascent algorithm and then switch to the quasi-Newton algorithm. Both algorithms and their combination are available in our Julia package.

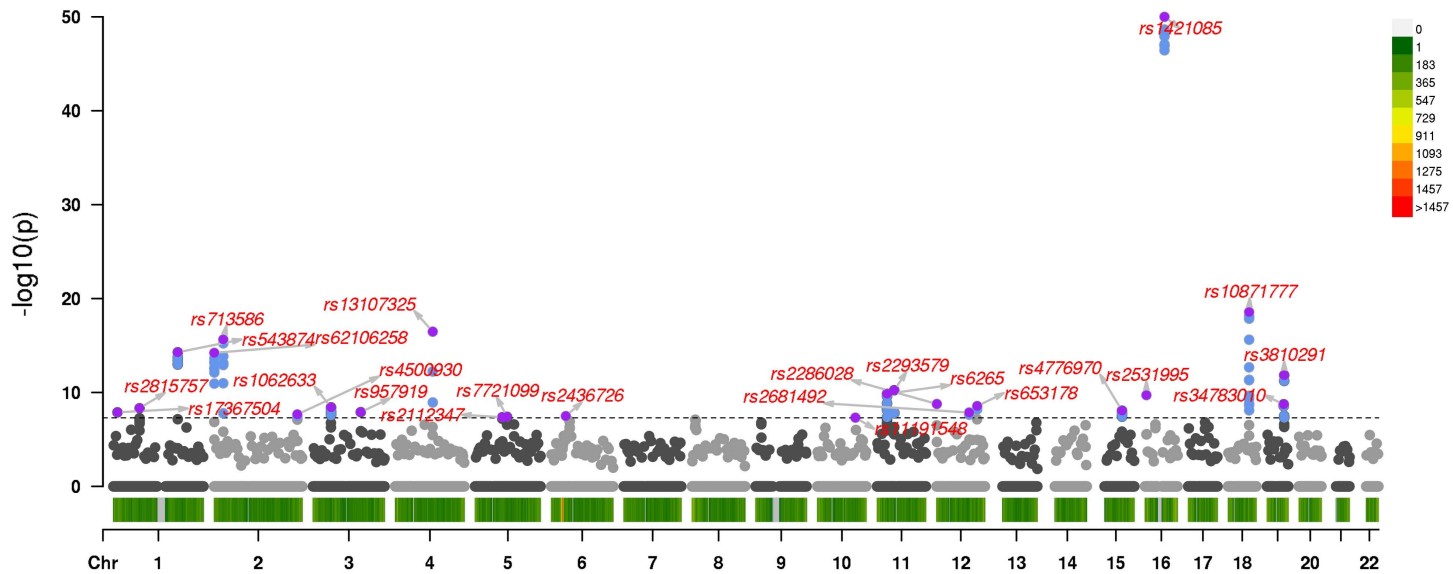

**Fig 3. A 3-trait multivariate GWAS on BMI, dichotomized SBP, and dichotomized DBP.** The black horizontal dotted line indicates the the genome-wide threshold of $5 \times 10^{-8}$. The most significant SNP within a 1Mb window is labeled and colored purple. All other significant SNPs are colored blue and unlabeled. The legend on the right shows chromosome density.

The approximate-copula model shines when GLMM and GEE are difficult to apply or yield unsatisfactory results. We illustrate this fact for longitudinal binary traits and multiple correlated traits of mixed types. To its credit, the approximate-copula model is likelihood-based, preserves base distributions approximately, and enables hypothesis testing of mean effects and correlations. However, our numerical tests suggest caution. In particular, performance may degrade when correlations between responses are strong or when cluster sizes are large. In this setting the normalizing constant $\ln[1 + \frac{1}{2} \text{tr}(\Gamma)]$ grows and tends to shrink estimates of $\Gamma$, making variance component estimates unreliable. In defense of the model, our limited evidence suggests that estimates of mean effects are hardly affected. We suspect that likelihood ratio tests of correlation are also reliable. Given the multivariate nature of the model, it also enhances statistical precision and power in the analysis of mean effects when correlations exist among the traits. In our opinion, variance components are nuisance parameters compared to mean effects in important examples such as GWAS.

In any case, we recommend univariate analysis by standard statistical tools as a preclude to multivariate analysis under the approximate-copula model. Understanding the nature of complicated data is best served when the data is approached from several angles. Despite its simplifications, the approximate-copula model's balance of speed, interpretability, and flexibility make it a valuable complement to existing methods. We hope that other statisticians will agree, work to improve its performance, and apply it to their own data.

### Web resources

Our software is freely available to the scientic community through the OpenMendel [22] platform.
**Project home page**: https://github.com/OpenMendel/ApproxCopula.jl
**Supported operating systems**: Mac OS, Linux, Windows
**Programming language**: Julia 1.6+
**License**: MIT
All commands needed to reproduce the following results are available at https://github.com/sarah-ji/ApproxCopula-reproducibility

## Supporting information

**S1 File. Additional mathematical derivations, simulations, as well as real data analysis results.**
(PDF)

## Acknowledgments

We thank Tetsuji Tonda for his pioneering contributions and Seyoon Ko for help in parallelizing our Julia code.

## Author contributions

**Conceptualization:** Hua Zhou, Kenneth Lange.

**Data curation:** Sarah S. Ji, Benjamin B. Chu.

**Formal analysis:** Sarah S. Ji, Benjamin B. Chu.

**Funding acquisition:** Hua Zhou, Kenneth Lange.

**Methodology:** Sarah S. Ji, Kenneth Lange.

**Project administration:** Kenneth Lange.

**Software:** Sarah S. Ji, Benjamin B. Chu, Hua Zhou.

**Supervision:** Hua Zhou, Kenneth Lange.

**Writing – original draft:** Sarah S. Ji, Benjamin B. Chu, Hua Zhou, Kenneth Lange.

**Writing – review & editing:** Sarah S. Ji, Benjamin B. Chu, Hua Zhou, Kenneth Lange.

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
