## [Editor Report · Decision Letter 0]

6 Feb 2025

PCOMPBIOL-D-25-00101

A Flexible Quasi-Copula Distribution for Statistical Modeling

PLOS Computational Biology

Dear Dr. Chu,

Thank you for submitting your manuscript to PLOS Computational Biology. As with all papers, your manuscript was reviewed by members of the editorial board. Based on our assessment, we have decided that the work does not meet our criteria for publication and will therefore be rejected.

We are sorry that we cannot be more positive on this occasion. We very much appreciate your wish to present your work in one of PLOS's Open Access publications. Thank you for your support, and we hope that you will consider PLOS Computational Biology for other submissions in the future.

Yours sincerely,

Michael B Sohn, Ph.D.

Guest Editor

PLOS Computational Biology

Ilya Ioshikhes

Section Editor

PLOS Computational Biology

**Additional Editor Comments (if provided):**

The paper should include essential details about the proposed distribution function. Specifically, it should address the following points: whether the function satisfies all the necessary conditions for quasi-copulas; whether Tonda's approximation is valid for univariate distributions that are mixtures from the exponential family; and what conditions must be fulfilled for the proposed distribution function to accurately approximate the true distribution.
---

## [Decision Letter · Decision Letter 1]

12 May 2025

PCOMPBIOL-D-25-00101R1

An Approximate-Copula Distribution for Statistical Modeling

PLOS Computational Biology

Dear Dr. Chu,

Thank you for submitting your manuscript to PLOS Computational Biology. Your manuscript has been reviewed by members of the editorial board and three independent reviewers. In light of the reviews (below this email), we would like to invite the resubmission of a significantly-revised version that takes into account the reviewers' comments.

The reviewers raised significant concerns regarding the proposed methodology and the benchmarking comparisons conducted. The authors need to carefully address each of the reviewers' comments. We cannot make any decision about publication until we have seen the revised manuscript and your response to the reviewers' comments. Your revised manuscript is also likely to be sent to reviewers for further evaluation.

Please submit your revised manuscript within 60 days Jul 06 2025 11:59PM. If you will need more time than this to complete your revisions, please reply to this message or contact the journal office at ploscompbiol@plos.org. Please include the following items when submitting your revised manuscript:

We look forward to receiving your revised manuscript.

Kind regards,

Michael B Sohn, Ph.D.

Guest Editor

PLOS Computational Biology

Ilya Ioshikhes

Section Editor

PLOS Computational Biology

**Journal Requirements:**

1) We have noticed that you have uploaded Supporting Information files, but you have not included a list of legends. Please add a full list of legends for your Supporting Information files after the references list.

2) Please include the grant recipients in the Funding Information tab.

3) Please provide a completed 'Competing Interests' statement in the online submission form including any COIs declared by your co-authors. If you have no competing interests to declare, please state "The authors have declared that no competing interests exist".

4) Thank you for stating that Sarah Ji and Soo Min Ji is the “same person.” Please confirm which name she would like to go by in publications.

**Reviewers' comments:**

Reviewer's Responses to Questions

**Comments to the Authors:**

**Please note that two reviews are uploaded as attachments.**

Reviewer #1: Please refer to the referee report.

Reviewer #2: See the attached report.

Reviewer #3: The authors proposed an approximate-copula model to model arbitrary correlation structures across different types of outcomes (continuous, binary, count). The marginal means of the approximate-copula model can be further linked with covariates to assess relationships between the multivariate outcomes and the covariates. I think this is a promising idea with potential applications in various fields, but I have a few major comments that I hope the authors can clarify:

1. The $\Gamma$ matrix appears to be the most important quantity in the proposed approximate-copula model. The authors refer to this matrix as “the target covariance matrix” in line 121, page 5. However, in line 141, page 7, the covariance between $Y_k$ and $Y_l$ is approximately equal to $\sigma_k \sigma_l \gamma_{kl}$. This suggests that $\Gamma$ is more akin to a correlation matrix rather than a covariance matrix. It would be helpful if the authors could clarify the interpretation and terminology used.

2. The moment calculations in Section 2.3 seem to rely on the assumption that the spectral norm of $\Gamma$ is small. If $\Gamma$ represents a correlation matrix, we can assume that all its diagonal entries are 1. In that case, the spectral norm of $\Gamma$ depends heavily on the off-diagonal entries—that is, the correlations between outcomes. In the simplest case, where all outcomes are uncorrelated, the spectral norm is 1. However, in extreme cases with strong correlations, the spectral norm can be quite large. This implies that assuming a small spectral norm may restrict the strength of allowable correlations between outcomes. In other words, when the outcomes are highly correlated, the marginal distribution of $Y$ could differ substantially from that of $X$.

3. Related to #2, if the marginal distribution of $Y$ differs considerably from that of $X$, then the approximate copula model may be modeling something different from the observed data. This could be particularly concerning in the regression step, especially in Equation (2), line 198, page 10, where the marginal mean of $X$ is linked to the covariates. This seems to present a gap, as the approximate copula model focuses on $Y$, while Equation (2) centers on $X$, especially when $X$ and $Y$ have different marginal distributions. This appears to be a potential model-specification issue.

4. In light of the issue raised in #3, it is unclear whether it is possible to construct valid confidence intervals for the MLEs derived in Section 2.9. Some clarification or justification would be appreciated.

5. (Minor) The multivariate GWAS application is very interesting. An alternative approach would be to conduct univariate GWAS for each outcome separately and then apply combination tests across outcomes (e.g., the Cauchy combination test). I would be curious to see whether the proposed multivariate GWAS approach offers any statistical power gains over these univariate strategies.

**Have the authors made all data and (if applicable) computational code underlying the findings in their manuscript fully available?**

Reviewer #1: Yes

Reviewer #2: Yes

Reviewer #3: None

PLOS authors have the option to publish the peer review history of their article (what does this mean? ). If published, this will include your full peer review and any attached files.

**Do you want your identity to be public for this peer review?** For information about this choice, including consent withdrawal, please see our Privacy Policy .

Reviewer #1: No

Reviewer #2: No

Reviewer #3: No

**Figure resubmission:**

**Reproducibility:**



---

## [Editor Report · Decision Letter 2]

4 Aug 2025

An Approximate-Copula Distribution for Statistical Modeling

PLOS Computational Biology

Dear Dr. Chu,

Thank you for submitting your manuscript to PLOS Computational Biology. In the previous review, three reviewers reviewed the submitted manuscript. However, you addressed the comments from only two of the reviewers. The authors are required to respond to all reviewers' comments thoroughly. Accordingly, we invite you to resubmit your manuscript, ensuring that you incorporate comments from all three reviewers in your revisions.

Please submit your revised manuscript within 60 days Oct 04 2025 11:59PM. If you will need more time than this to complete your revisions, please reply to this message or contact the journal office at ploscompbiol@plos.org. Please include the following items when submitting your revised manuscript:

We look forward to receiving your revised manuscript.

Kind regards,

Michael B Sohn, Ph.D.

Guest Editor

PLOS Computational Biology

Ilya Ioshikhes

Section Editor

PLOS Computational Biology

**Journal Requirements:**

1) Please provide an Author Summary. This should appear in your manuscript between the Abstract (if applicable) and the Introduction, and should be 150-200 words long. The aim should be to make your findings accessible to a wide audience that includes both scientists and non-scientists. Sample summaries can be found on our website under Submission Guidelines:

2) We have noticed that you have uploaded Supporting Information files, but you have not included a list of legends. Please add a full list of legends for your Supporting Information files after the references list.

3) Please provide a completed 'Competing Interests' statement, including any COIs declared by your co-authors. If you have no competing interests to declare, please state "The authors have declared that no competing interests exist".

4) Please ensure that the funders and grant numbers match between the Financial Disclosure field and the Funding Information tab in your submission form. Note that the funders must be provided in the same order in both places as well. Currently, the order of this grant "HG006139" is different in both places. Please also ensure that the recipients are included in the Funding Information tab.

**Figure resubmission:**

**Reproducibility:**



---

## [Decision Letter · Decision Letter 3]

22 Oct 2025

An Approximate-Copula Distribution for Statistical Modeling

PLOS Computational Biology

Dear Dr. Chu,

Thank you for submitting your manuscript to PLOS Computational Biology. After careful consideration, we feel that it has merit but does not fully meet PLOS Computational Biology's publication criteria as it currently stands. Therefore, we invite you to submit a revised version of the manuscript that addresses the points raised during the review process.

Please submit your revised manuscript within 60 days Dec 22 2025 11:59PM. If you will need more time than this to complete your revisions, please reply to this message or contact the journal office at ploscompbiol@plos.org. Please include the following items when submitting your revised manuscript:

We look forward to receiving your revised manuscript.

Kind regards,

Michael B Sohn, Ph.D.

Guest Editor

PLOS Computational Biology

Ilya Ioshikhes

Section Editor

PLOS Computational Biology

**Journal Requirements:**

1) Please provide an Author Summary. This should appear in your manuscript between the Abstract (if applicable) and the Introduction, and should be 150-200 words long. The aim should be to make your findings accessible to a wide audience that includes both scientists and non-scientists. Sample summaries can be found on our website under Submission Guidelines:

2) We have noticed that you have uploaded Supporting Information files, but you have not included a list of legends. Please add a full list of legends for your Supporting Information files after the references list.

Please check the citations of the supplementary files within the text of the manuscript and match the names of your supporting information files with the supporting information captions within your manuscript

Note: Authors may use almost any description as the item name for a supporting information file as long as it contains an “S” and number. For example, “S1 Appendix” and “S2 Appendix,” “S1 Table” and “S2 Table,” and so forth.  .

3) Please ensure that the funders and grant numbers match between the Financial Disclosure field and the Funding Information tab in your submission form. Note that the funders must be provided in the same order in both places as well. Currently, the order of this grant "HG006139" does not match in both places.

**Reviewers' comments:**

Reviewer's Responses to Questions

Reviewer #1: The authors have addressed all my previous comments.

Reviewer #2: I have no further comments.

Reviewer #3: 1. In response to Comment 1, the authors wrote that “the entry $\gamma_{ij}$ is neither the covariance nor the correlation between $Y_i$ and $Y_j$.” However, in response to Comment 2, they stated that “as mentioned in the response to the previous question, $\Gamma$ is a covariance matrix.” Could the authors please clarify this?

2. I appreciate the clarification regarding the approximate copula model and its impact on preserving the marginals of $X$. However, it remains unclear how this approximation might affect the overall performance of the proposed method. The authors mention that the approach tends to perform well when correlations between outcomes are weak (as noted in their response to Comment 3: “this assumption may also be reasonable, especially when the observed correlation is weak”). Could the authors elaborate on what this means in practice? For instance, should practitioners consider using this method primarily when correlations are weak? The current presentation is mathematically detailed, but it would greatly benefit from clearer guidance for applied researchers regarding the strengths and limitations of the method.

3. I appreciate the authors’ comparison with univariate GWAS and their careful interpretation of results. However, from a practitioner’s perspective, it is still unclear why one should choose this method over the simpler univariate GWAS approach, particularly if it may not provide higher power or stronger biological insights. It would be helpful if the authors could articulate more concretely the advantages of the proposed approach in real applications. PLOS Computational Biology values methodological innovation that provides meaningful biological or practical impact; without such clarification, the application section appears as a technical illustration rather than a demonstration of real-world utility.

4. Finally, the authors should carefully review the revised manuscript, as there are still several places where question marks remain in the text.

**Have the authors made all data and (if applicable) computational code underlying the findings in their manuscript fully available?**

Reviewer #1: Yes

Reviewer #2: None

Reviewer #3: None

PLOS authors have the option to publish the peer review history of their article (what does this mean? ). If published, this will include your full peer review and any attached files.

**Do you want your identity to be public for this peer review?** For information about this choice, including consent withdrawal, please see our Privacy Policy .

Reviewer #1: No

Reviewer #2: No

Reviewer #3: No

**Figure resubmission:**
---

## [Editor Report · Decision Letter 4]

5 Jan 2026

PCOMPBIOL-D-25-00101R4

An Approximate-Copula Distribution for Statistical Modeling

PLOS Computational Biology

Dear Dr. Chu,

Thank you for submitting your manuscript to PLOS Computational Biology. After careful consideration, we feel that it has merit but does not fully meet PLOS Computational Biology's publication criteria as it currently stands. Therefore, we invite you to submit a revised version of the manuscript that addresses the points raised during the review process.

We look forward to receiving your revised manuscript.

Kind regards,

Michael B Sohn, Ph.D.

Guest Editor

PLOS Computational Biology

Ilya Ioshikhes

Section Editor

PLOS Computational Biology

**Additional Editor Comments:**

The definition of Gamma is inconsistent throughout the paper and the authors' responses. The authors stated that Gamma is neither a covariance nor a correlation matrix. However, Gamma_i in sections 2.6, 2.7, 2.9, and 3.1 are stated and defined as a covariance matrix. Please clarify what Gamma is and use the term consistently throughout the paper. Also, what does m indicate in the expression of Gamma_i on page 10, i.e., Gamma_i = sum_{j=1}^m theta_i Omega_{ij}?

**Reviewers' comments:**

**Figure resubmission:**
---

## [Editor Report · Decision Letter 5]

16 Jan 2026

Dear Mr Chu,

We are pleased to inform you that your manuscript 'An Approximate-Copula Distribution for Statistical Modeling' has been provisionally accepted for publication in PLOS Computational Biology.

Best regards,

Michael B Sohn, Ph.D.

Guest Editor

PLOS Computational Biology

Ilya Ioshikhes

Section Editor

PLOS Computational Biology

---

## [Editor Report · Acceptance letter]

PCOMPBIOL-D-25-00101R5

An Approximate-Copula Distribution for Statistical Modeling

Dear Dr Chu,

I am pleased to inform you that your manuscript has been formally accepted for publication in PLOS Computational Biology. Your manuscript is now with our production department and you will be notified of the publication date in due course.

With kind regards,

Judit Kozma
